# Prognostic role of PD-L1 expression in patients with salivary gland carcinoma: A systematic review and meta-analysis

Limeng Wu[1], Canhua Jiang[2,3], Zhihui Zhu[1], Yao Sun[4], Tao Zhang[1]*

**1** Department of Stomatology, Peking Union Medical College Hospital, Peking Union Medical College, Chinese Academy of Medical Sciences, Beijing, People's Republic of China, **2** Department of Oral and Maxillofacial Surgery, Center of Stomatology, Xiangya Hospital, Central South University, Changsha, Hunan, People's Republic of China, **3** Research Center of Oral and Maxillofacial Tumor, Xiangya Hospital, Central South University, Changsha, Hunan, People's Republic of China, **4** Department of Radiation Oncology, CyberKnife Center, Key Laboratory of Cancer Prevention and Therapy, Tianjin Medical University Cancer Institute and Hospital, National Clinical Research Center for Cancer, Tianjin, People's Republic of China

\* drtzhang@126.com

## Abstract

### Background

Although programmed cell death-ligand 1 (PD-L1) has been recognized as a potential marker in several cancers, the relationship between PD-L1 expression and survival in patients with salivary gland carcinoma (SGC) has remained unclear. We aimed to evaluate the association of PD-L1 expression with clinicopathological features and prognosis in SGC patients.

### Methods

The databases Ovid Medline, PubMed, Scopus, and EMBASE were searched for relevant studies that detected PD-L1 expression in SGC. The meta-analysis was performed according to the preferred reporting items for systematic reviews and meta-analyses (PRISMA), and the reporting recommendations for tumor marker prognostic studies (REMARK) was used to assess the quality of research eligible for this meta-analysis. Included studies were assessed using the Quality in Prognosis Studies (QUIPS) tool. Odds ratios (ORs) with 95% confidence interval (CI) were calculated to estimate the correlation between PD-L1 expression and clinicopathological features. Hazard ratios (HRs) with 95% CI were applied to assess the association between PD-L1 expression and survival outcomes of patients.

### Results

A total of ten studies (including 952 patients with SGC) were evaluated. The meta-analysis showed that positive PD-L1 expression in SGC was significantly associated with male patients, older age, Tumor stage, lymph node metastasis, high pathological grade, and non-adenoid cystic carcinoma subtype. The pooled data demonstrated that high PD-L1 expression was associated with poor overall survival and disease-free survival. There was no

**Data Availability Statement:** All relevant data are within the paper and its Supporting Information files.

**Funding:** The author(s) received no specific funding for this work.

**Competing interests:** The authors have declared that no competing interests exist.

significant correlation between PD-L1 expression and progression-free survival or disease-specific survival of SGC patients.

## Conclusion

According to the meta-analysis, positive PD-L1 expression may play an important role as an effective marker of poor prognosis in patients with SGC. However, large-scale, prospective investigations are still needed to confirm the findings. The assessment of PD-L1 expression may aid in the personalized management of SGC.

## Introduction

Salivary gland carcinomas (SGCs), with an incidence of 12–30 per one million people, are relatively rare malignancies, accounting for more than 0.5% of all malignancies and approximately 1%–8% of head and neck carcinomas worldwide [1–5]. SGCs presented with strong heterogeneity, which leads to different histological subtypes [6], there are more than 20 different pathological subtypes according to the World Health Organization. Thus, complex biological behaviors of SGCs presented greater differences in the clinical course. SGCs are generally slow-growing and characterized by frequent local recurrence and latent distant metastases. Surgical resection is the standard treatment for localized SGC [7], radiation and/or chemotherapy are considered in the adjuvant setting. However, the 5-year postoperative recurrence rate ranged from 25 to 46% [5, 8–10]. Predicting the outcome of SGC patients is challenging for clinicians due to the diversity of tumor behavior.

The programmed cell death ligand (PD-L1) is expressed in resting immune cells and various tumor cells. When the programmed death-1 (PD-1) molecule binds to the PD-L1 receptor ligand, a complex is formed that suppresses cytotoxic T cells, resulting in apoptosis and tumor escape. Overexpression of PD-L1 has been linked to a poor prognosis and outcome in malignant tumors such as breast cancer, gastrointestinal tract cancer, cholangiocarcinoma, and renal cell carcinoma [11–15]. However, the role of PD-L1 expression in SGC remains unclear. Several cohort studies showed that positive expression of PD-L1 was associated with poor survival of SGC patients [16–18]. However, other studies reported different outcomes [19, 20]. Since controversial conclusions remain, it is necessary to analyze to clarify the role of PD-L1 in SGC patients and obtain a universal conclusion to guide clinical practice.

In this study, we conducted a systematic review and meta-analysis to assess the relationships between PD-L1 expression and clinicopathological features of SGC patients, as well as the association between PD-L1 expression and SGC patient survival outcomes.

## Materials and methods

### Search protocol

A comprehensive literature search was conducted for potential articles on Ovid Medline, PubMed, Scopus, and EMBASE. The search strategies were based on combinations of the following keywords: "PD-L1, B7-H1, CD274, programmed cell death ligand 1", "salivary gland, parotid gland, submandibular gland, sublingual gland, salivary duct, adenoid cystic, mucoepidermoid", "tumor, neoplasm, cancer, carcinoma", and "survival, prognostic, prognosis, outcome, mortality". The search was limited to articles in the English language. Both published and unpublished studies were sought. The endpoint of the search was January 13, 2022. The

search was re-run before the final analysis. This meta-analysis was performed based on the Preferred Reporting Items for Systematic Review and Meta-Analysis (PRISMA) guidelines [21] and the reporting recommendations for tumor marker prognostic studies (REMARK) [22] were followed when we conducted and reported this meta-analysis. The proposal was registered a priori at the National Institute for Health Research, International Prospective Register of Systematic Reviews (PROSPERO; name: prognostic role of PD-L1 expression in patients with salivary gland carcinoma: a systematic review and meta-analysis; registration number: CRD42020183228). The review protocol was not published.

## Selection of studies

Studies were considered eligible for inclusion if they satisfied the following criteria: (1) clinical cohort studies; (2) PD-L1 expression in the tumor tissues was detected by immunohistochemistry (IHC); (3) investigated the PD-L1 expression of SGC patients, included both positive and negative expression patients; (4) the relationships between PD-L1 expression and clinico-pathological features, prognosis outcomes were depicted in the study; (5) provided information on the relationships of PD-L1 expression with survival parameters in patients with SGC or sufficient information for estimating the hazard ratio (HR).

Exclusion criteria were as follows: (1) studies reported in conference abstracts, reviews, or letters; (2) studies with insufficient data, no access to the full manuscript, reported published data, and non-English language publications.

## Data extraction and quality assessment

Two reviewers separately screened the titles and abstracts of publications identified by the search strategy to exclude irrelevant research. Any discrepancies between the two reviewers were resolved by discussion and consensus. The two reviewers carefully examined the full manuscripts of potentially eligible articles to determine whether they should be included or not. Disagreements were resolved by discussion or consultation with a third reviewer. Microsoft Excel spreadsheet was used to record decisions. Data extraction was independently performed by 2 reviewers. Basic information of the studies such as first author, publication year, the origin of population, number of patients, and clinical and pathological characteristics of patients such as gender, age, subtype of SGC, TNM category, tumor grade, clinical stage, perineural invasion (PNI), vascular invasion (VI), margin status, treatment outcome, type of tissue for PD-L1 expression analysis, the cut-off value for PD-L1 positivity, and PD-L1 expression score were extracted.

Patient survival outcomes included overall survival (OS), disease-free survival (DFS), progression-free survival (PFS), and disease-specific survival (DSS). OS is defined as the time interval between the date of primary treatment and the date of death or last follow-up; DFS is defined as the period of time without evidence of disease (i.e. persistence, recurrence or metastasis) after primary treatment; DSS is defined as the duration after primary treatment to death due to SGC; PFS is defined as the length of time during and after the primary treatment of SGC that the patient lives with the disease but it does not get worse. HR and the corresponding 95% confidential interval (CI) for patient survival data were extracted. For studies that showed survival data indirectly with a Kaplan-Meier curve, the methods utilized were described by Tierney et al. [23] to estimate. Disagreements were resolved by discussion or consultation with a third reviewer. Microsoft Excel spreadsheet was used to record data.

## Risk of bias

Two reviewers independently assessed included studies for risk of bias using the Quality In Prognosis Studies (QUIPS) tool [24]. The QUIPS tool contains six domains: study

participation, study attrition, prognostic factor measurement, outcome measurement, study confounding, statistical analysis and reporting. Each domain contains 3–7 prompting items and considerations, and each domain is rated as high, moderate, or low risk of bias considering the prompting items. The overall risk of bias for each study was categorized as low if the risk of bias was low in all domains, moderate if the risk of bias was deemed to have moderate in one to four domains, or high if the risk of bias was high in at least one domain. Any discrepancies were resolved by discussion and consensus.

## Statistical analysis

In this meta-analysis, all extracted data were processed using Stata software (version 15.0; Stata Corp LP, College Station, TX), R Studio (Version 1.2.1335; R Studio, Boston, MA). The relationships between PD-L1 expression and clinical and pathological parameters were evaluated using an odds ratio (OR) with 95% Cls. HR with 95% CI was used to assess the association between PD-L1 expression and the survival of SGC patients. Statistical heterogeneity among studies was assessed via the $\chi^2$ test and the $I^2$ test. We applied a random-effects model to calculate the pooled data if significant heterogeneity is detected ($P<0.1$ or $I^2>50\%$). Otherwise, a fixed-effects model was implemented. A $P$-value less than 0.05 was considered statistically significant.

For the heterogeneity of SGC (both subtype of SGCs and heterogeneity within one tumor tissue), different detection techniques, and cutoff values of PD-L1 expression, subgroup investigations were performed. Subgroup analysis, stratified by sample size (0-100/>100), the cutoff value for PD-L1 positivity (1%/5%), PD-L1 expression score (tumor proportion score/combined positivity score, TPS/CPS), tumor tissue used for PD-L1 express detection (whole tissue section/tissue microarrays), and HR estimation (reported/calculated), were performed. We applied a random-effects model to calculate the pooled data if significant heterogeneity is detected ($P<0.1$ or $I^2>50\%$). Otherwise, a fixed-effects model will be implemented. A $P$-value less than 0.05 was considered statistically significant. The sensitivity analyses were also conducted to assess the robustness of the results and to explore the impact of each single study on the overall effect.

## Results

### Search results

The process of literature search and study selection is presented in Fig 1. A total of 207 studies were retrieved from searches of databases, and 121 studies were excluded as duplicates. A total of 48 studies had evaluated PD-L1 expression in SGC. After examining the title and abstract, 14 reports were discarded. Subsequently, the full text of the remaining 34 studies was evaluated, and 24 studies were excluded owing to the following reasons: six studies were conference abstracts or case reports, five studies did not evaluate the association between PD-L1 expression and clinicopathological data [20, 25–28], eight studies did not provide survival information [29–36], four studies did not provide sufficient data for HR calculation [18, 37–39], and 1 study was overlapped [40]. The present meta-analysis includes 10 studies with 952 patients which matched the eligibility criteria [16, 17, 19, 41–47].

### Study characteristics

A total of 952 patients from 10 independent cohorts were diagnosed with SGC, with various primary site including major and minor salivary glands. All the included studies were retrospective and published between 2016 and 2021. The patient number of the studies ranged

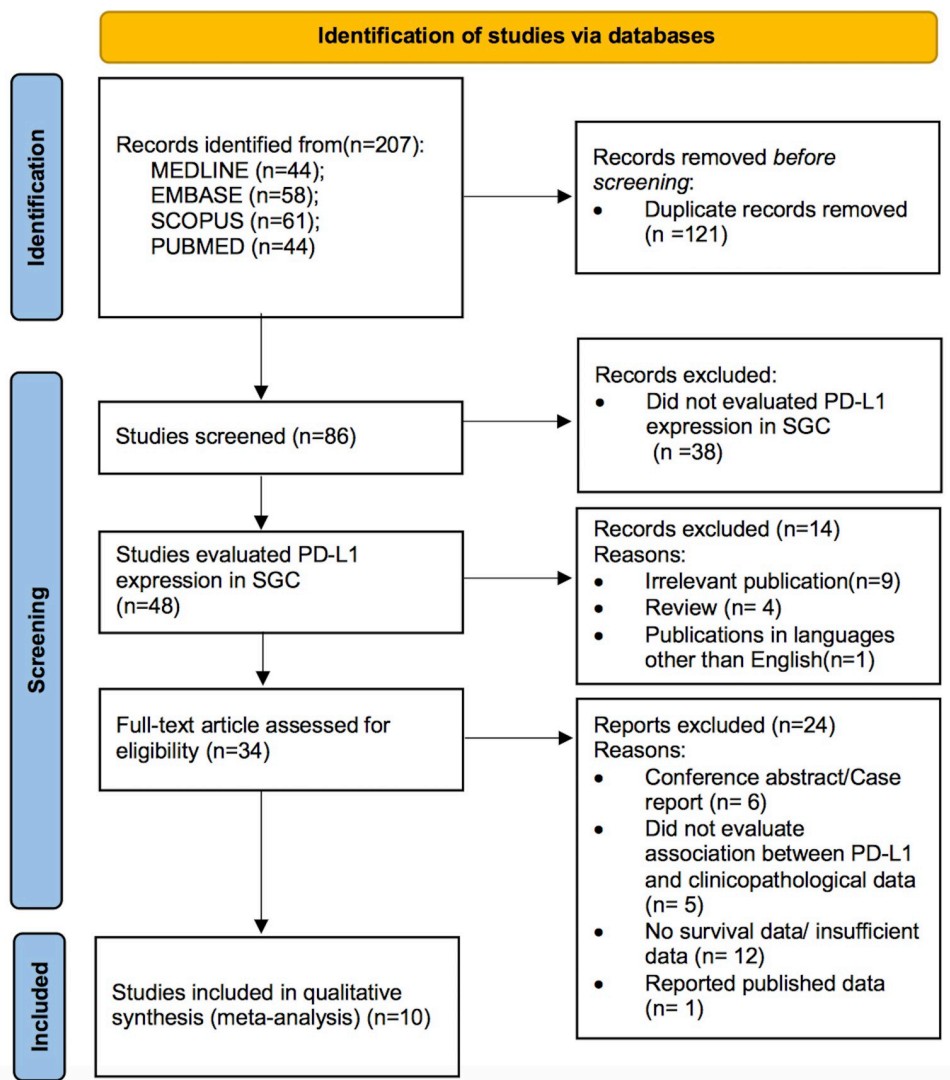

**Fig 1. Flow chart of literature search and study selection.**

from 30 to 219. The patients were enrolled from 5 different countries: Poland [19], Japan [16, 17, 41, 44, 45], Germany [42, 47], Australia [43], and China [46].

IHC was applied to detect the expression of the PD-L1 protein in all studies. The definitions of positive staining of PD-L1 were different among studies since there has been no consensus so far. In the studies by Szewczyk [19], Nakano [45], Higashino [41], Kesar [42], only the positive IHC stains in tumor cells were deemed valid. In the other five studies, positive stains in tumor cells and tumor-infiltrating mononuclear cells (TIMCs) were presented [16, 17, 44, 46, 47], and 4 studies counted both tumor cells and TIMCs as one of the positive staining [43, 44, 46, 47], and therefore, survival data from studies applying CPS and immune cells (ICs) score for positive PD-L1 expression were also pooled to investigate the prognostic roles. PD-L1 expression was defined positively when positive cells were with membranous and/or cytoplasmic staining. The cut-off value of PD-L1 positivity was 1% in seven studies [16, 41, 43–47], 5% in three studies [17, 19, 42].

**Table 1. Summary of the studies that examined the prognostic value of PD-L1 in SGC.**

| Author | Year | Country | No. of patients | The cut-off value for PD-L1 positive | No. of positive expression n (%) | Follow-up time (months) | Adjuvant treatment | Survival parameter |
|---|---|---|---|---|---|---|---|---|
| Szewczyk | 2019 | Poland | 117 | >5% TC | 22 (18.8) | 31 (2–110) | RT | DFS[U] |
| Nakano | 2019 | Japan | 30 | >1% TC | 11 (36.7) | NA | NA | DFS[M], DSS[M] |
| Harada | 2018 | Japan | 47 | >5% TC | 24 (51.1) | 88.8 (22–154) | NA | OS[U] |
| | | | | >5% TIMC | 20 (42.6) | | | OS[U] |
| Mukaigawa | 2016 | Japan | 219 | >1% TC | 50 (22.8) | 44.9 (0.4–221) | NA | OS[M], DFS[M] |
| | | | | >1% TIMC | 28 (12.8) | | | OS[M], DFS[M] |
| Witte | 2020 | Germany | 94 | >1% TPS | 61 (64.9) | 89.5 (12–240) | RT, CHEMO | PFS[U], OS[U] |
| | | | | >1% CPS | 75 (79.8) | | | PFS[U], OS[U] |
| | | | | >5% IC [#] | 25 (26.9) | | | PFS[#], OS[#] |
| Higashino | 2020 | Japan | 127 | >1% TC | 36 (28.3) | 38 (2–213) | RT | DSS[U] |
| Sato | 2021 | Japan | 73 | >1% TPS | 31 (42.0) | 64.1 (1.9–205) | NA | DFS[U, M], OS[U, M] |
| | | | | >1% MIDS | 44 (60.0) | | | DFS[U], OS[U] |
| | | | | >1% CPS | 46 (63.0) | | | DFS[U], OS[U] |
| Kesar | 2020 | Germany | 84 | >5% TPS | 14 (17.5) | 55 (0–443) | RT, CHEMO | OS[U,M], PFS[U,M] |
| Guazzo | 2021 | Australia | 52 | >1% CPS | 9 (17.3) | 52.3 (0.7–123) | RT, CHEMO | OS[U] |
| Fang | 2021 | China | 109 | >1% TPS | 47 (43.1) | 45.6 (7–115) | RT, CHEMO | OS[U, M], DFS[U, M] |
| | | | | >1% CPS | 87(79.8) | | | OS[U], DFS[U] |
| | | | | >1% IC | 48(44.0) | | | OS[U], DFS[U] |

[#]: survival data were not available for HR estimation; TC, tumor cell; RT, radio therapy; DFS, disease-free survival; NA, not available; DSS, disease-specific survival; OS, overall survival; TIMC, tumor-infiltrating mononuclear cell; TPS, tumor proportion score; CHEMO, chemotherapy; U, univariate analysis; M, multivariate analysis; PFS, progression-free survival; CPS, combined positivity score; IC, immune cell.

Regarding the prognosis of SGC patients, seven studies provided data about the association between PD-L1 and OS [16, 17, 42–44, 46, 47], five studies provided data on the association between PD-L1 expression and DFS [16, 19, 44–46], two studies provided data on the association between PD-L1 expression and PFS [42, 47], and two studies provided data on the association between PD-L1 expression and DSS [41, 45]. More detailed information of the eligible studies was presented in Table 1.

## PD-L1 expression and clinicopathological features

PD-L1 overexpression was associated with male patients, older age, T stage, positive N, higher pathological grade, and non-adenoid cystic carcinoma (ACC) subtype (Table 2). There was no statistically significant correlation between PD-L1 expression and Clinical Stage, PNI, VI, surgical margin, or treatment failure (Table 2).

## Risk of bias in studies

The 10 included studies were assessed using the QUIPS tool (Fig 2). Three studies were deemed to have a low risk of bias for all domains [42, 44, 46]. The other studies were rated with "moderate risk" [16, 17, 19, 41, 43, 45, 47]. As to the study participation domain, studies with consecutive samples and adequate sample size were considered "low risk". One manuscript was judged to be "moderate risk" due to representative cases selection for both tumor and normal tissues collection for further analyses [45]. In assessing the bias of study attrition domain, studies with complete follow-up data were identified as having "low risk of bias". Studies with a clear description or description of positive PD-L1 expression, tumor tissue

**Table 2. Meta-analysis of PD-L1 expression and clinicopathological features in SGC patients.**

| Clinical parameters | No. of studies (No. of patients) | OR (95%CI) | Model | Heterogeneity | | Significance(P) |
|---|---|---|---|---|---|---|
| | | | | $I^2$ | P | |
| Gender (Male&Female) | 7(614) [16, 17, 42–46] | 1.735(1.194–2.521) | Fixed | 27.50% | 0.219 | 0.004 |
| Age (Old&Young)[a] | 7(614) [16, 17, 42–46] | 1.536(1.063–2.218) | Fixed | 28.20% | 0.213 | 0.022 |
| T stage (T3T4&T1T2) | 8(811) [16, 19, 41–46] | 1.829(1.019–3.283) | Random | 56.60% | 0.024 | 0.043 |
| N stage (N+&N-) | 7(694) [16, 41–46] | 3.263(2.276–4.679) | Fixed | 0.00% | 0.462 | 0.000 |
| Clinical Stage (III+IV&I+II) | 2(120) [17, 44] | 2.115(0.720–6.212) | Random | 48.90% | 0.162 | 0.173 |
| Pathological Grade (H&L+M) | 4(449) [16, 41, 44, 45] | 7.248(4.335–12.118) | Fixed | 20.70% | 0.286 | 0.000 |
| PNI (P&N) | 4(453) [16, 43, 44, 46] | 1.460(0.596–3.577) | Random | 70.70% | 0.017 | 0.408 |
| VI (P&N) | 4(453) [16, 43, 44, 46] | 2.812(0.940–8.407) | Random | 78.80% | 0.003 | 0.064 |
| Surgical margin (P&N) | 4(272) [19, 43–45] | 1.043(0.577–1.885) | Fixed | 10.10% | 0.343 | 0.889 |
| Subtype (non-ACC&ACC) | 5(554) [16, 41, 42, 45, 47] | 7.714(1.509–39.442) | Random | 71.30% | 0.008 | 0.014 |
| Treatment Failure (P&N) | 3(194) [17, 19, 45] | 1.417(0.360–5.571) | Random | 72% | 0.028 | 0.618 |

[a]: Cutoff value of age included 45,60,61,63,64,65 and 66 years old, according to the median age of the included studies; OR, odds ratio; T stage, tumor stage; N stage, node stage; H, high; L, low; M, medium; PNI, perineural invasion; VI, vascular invasion; N, negative; P, positive; ACC, adenoid cystic carcinoma

utilized for PD-L1 express detection (tissue microarrays/ entire tissue slice), and cut-off values were graded as having a low risk of bias for prognostic factor measurement. A clear definition of the survival outcomes was considered as low risk of bias for the outcome measurement domain. Four studies presented both univariable and multivariable outcome [16, 42, 44, 46], one study provided the latter [45], these studies were rated as having low risk of bias for study confounding domain. For statistical analysis and reporting, direct presentation of data for survival outcome was rated as "low risk".

## PD-L1 expression and survival outcomes

**Association between PD-L1 expression and OS of SGC patients.** Data on the relationship between PD-L1 expression and OS were extracted from seven studies including 678 patients [16, 17, 42–44, 46, 47]. The heterogeneity was not significant ($I^2$ = 0.0%, $p$ = 0.814); therefore, a fixed-effects model was adopted. As shown in Fig 3A, the pooled data demonstrated that there was a significant correlation between PD-L1 expression and OS of SGC patients (n = 7, HR = 1.587, 95% CI = 1.175–2.144, $P$ = 0.003). Subgroup analysis was carried out based on sample size, cut-off value, PD-L1 expression score, PD-L1 expression type, and HR estimation. The data showed that PD-L1 is a significant factor of poor OS in SGC patients in studies that included less than a hundred patients, defining PD-L1 positive as cut-off value >1, using TPS as PD-L1 expression, utilized whole tissue section for the investigation. Whereas meta-regression analysis showed that no subgroups contribute to the heterogeneity of OS (Table 3).

Four included studies presented data with PD-L1 expression based on CPS score and OS [43, 44, 46, 47]. The heterogeneity was not significant ($I^2$ = 0.0%, $p$ = 0.908); therefore, a fixed-effects model was adopted. The pooled data revealed that there was no correlation between CPS score and OS in SGC patients (n = 4, HR = 0.592, 95% CI = 0.343–1.022, $P$ = 0.060, Fig 4A). Another four studies presented data with PD-L1 expression based on IC score and OS [16, 17, 44, 46]. The heterogeneity was not significant ($I^2$ = 0.0%, $p$ = 0.438); therefore, a fixed-effects model was adopted. The pooled data revealed that there was no correlation between IC score and OS in SGC patients (n = 4, HR = 1.068, 95% CI = 0.702–1.626, $P$ = 0.758, Fig 4B).

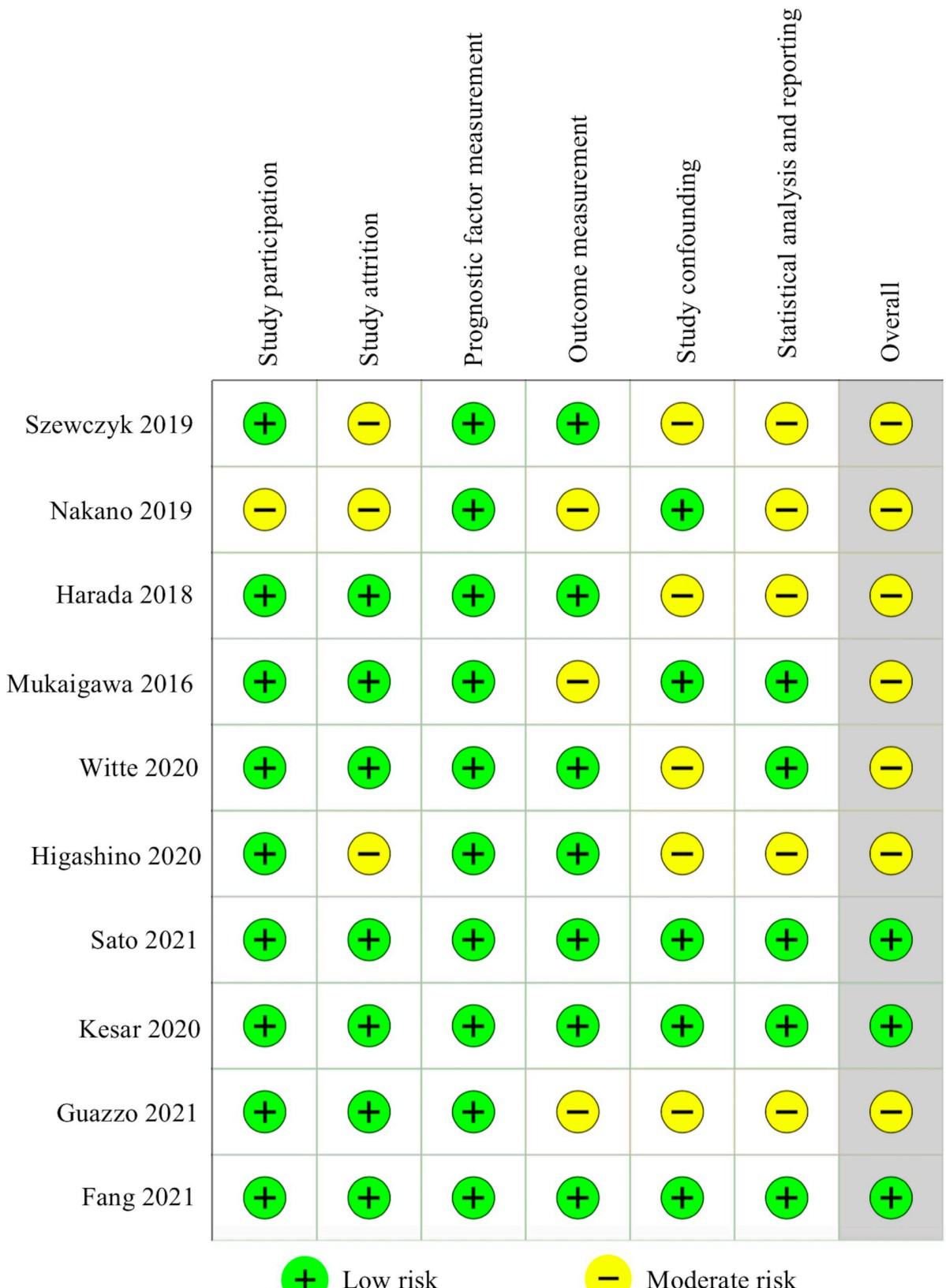

**Fig 2. Risk of bias assessment using the Quality In Prognosis Studies (QUIPS) tool.**

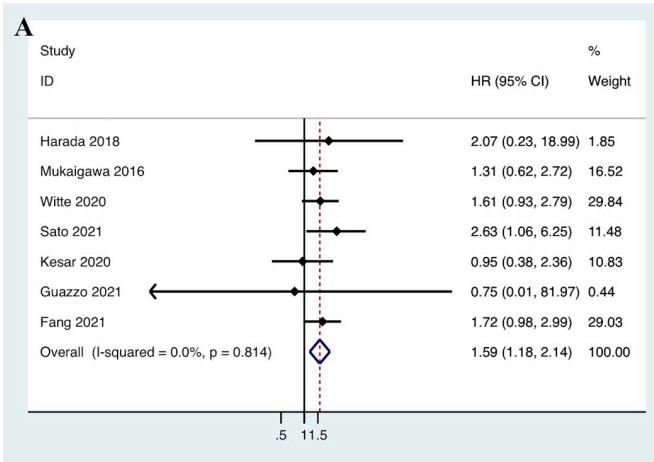
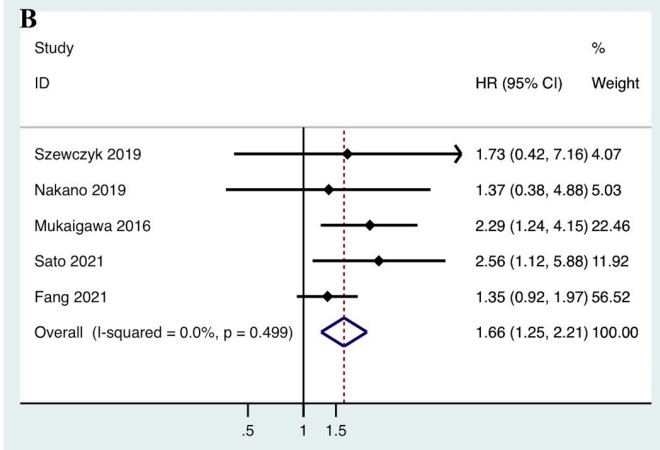

**Fig 3.** Forest plot for the association between programmed cell death ligand 1 (PD-L1) expression and (A) the overall survival (OS); and (B) the disease-free survival (DFS) of salivary gland carcinoma (SGC) patients. HR, hazard ratio; CI: confidence interval.

## Association between PD-L1 expression and DFS of SGC patients

Five studies with 548 patients provided the HRs for DFS [16, 19, 44–46]. The heterogeneity was not significant ($I^2$ = 0.0%, $P$ = 0.499); therefore, a fixed-effects model was applied. The pooled HR demonstrated a significant correlation between PD-L1 expression and DFS (n = 5, HR = 1.658, 95% CI = 1.246–2.208, $P$ = 0.001; Fig 3B).

Two included studies presented data with PD-L1 expression based on CPS score and DFS [44, 46]. The heterogeneity was not significant ($I^2$ = 0.0%, $p$ = 0.780); therefore, a fixed-effects model was adopted. The pooled data showed that there was no correlation between CPS score and DFS in SGC patients (n = 2, HR = 0.707, 95% CI = 0.409–1.221, $P$ = 0.213, Fig 4C). Three studies

**Table 3.** Subgroup analyses of survival outcomes based on different factors.

| Subgroup | No. of studies (No. of patients) | HR (95%CI) | Model | Heterogeneity | | Significance (P) | Meta-regression (P) |
|---|---|---|---|---|---|---|---|
| | | | | $I^2$ | $P$ | | |
| Total | 7 (678) | 1.587(1.175–2.144) | Fixed | 0.00% | 0.814 | 0.003 | |
| Sample size | | | | | | | 0.913 |
| >100 | 2 (328) | 1.558 (0.998–2.432) | Fixed | 0.00% | 0.564 | 0.051 | |
| <100 | 5 (350) | 1.612 (1.073–2.422) | Fixed | 0.00% | 0.624 | 0.022 | |
| Cut-off value | | | | | | | 0.321 |
| >1% | 5 (547) | 1.682 (1.219–2.320) | Fixed | 0.00% | 0.815 | 0.002 | |
| >5% | 2 (131) | 1.065 (0.458–2.475) | Fixed | 0.00% | 0.523 | 0.884 | |
| PD-L1 expression score | | | | | | | 0.744 |
| TPS | 6 (626) | 1.593 (1.178–2.152) | Fixed | 0.00% | 0.722 | 0.002 | |
| CPS | 1 (52) | 0.750 (0.008–67.903) | - | - | - | 0.9 | |
| Type of PD-L1 expression | | | | | | | 0.578 |
| tissue microarrays | 1 (219) | 1.310 (0.625–2.744) | - | - | - | 0.474 | |
| whole tissue section | 6 (459) | 1.649 (1.187–2.291) | Fixed | 0.00% | 0.753 | 0.003 | |
| HR estimation | | | | | | | 0.726 |
| calculated | 3 (278) | 1.719 (1.005–2.940) | Fixed | 0.00% | 0.924 | 0.048 | |
| reported | 4 (400) | 1.531 (1.065–2.200) | Fixed | 0.00% | 0.443 | 0.021 | |

HR, hazard ratio; TPS, tumor proportion score; CPS, combined positivity score; CI: confidence interval.

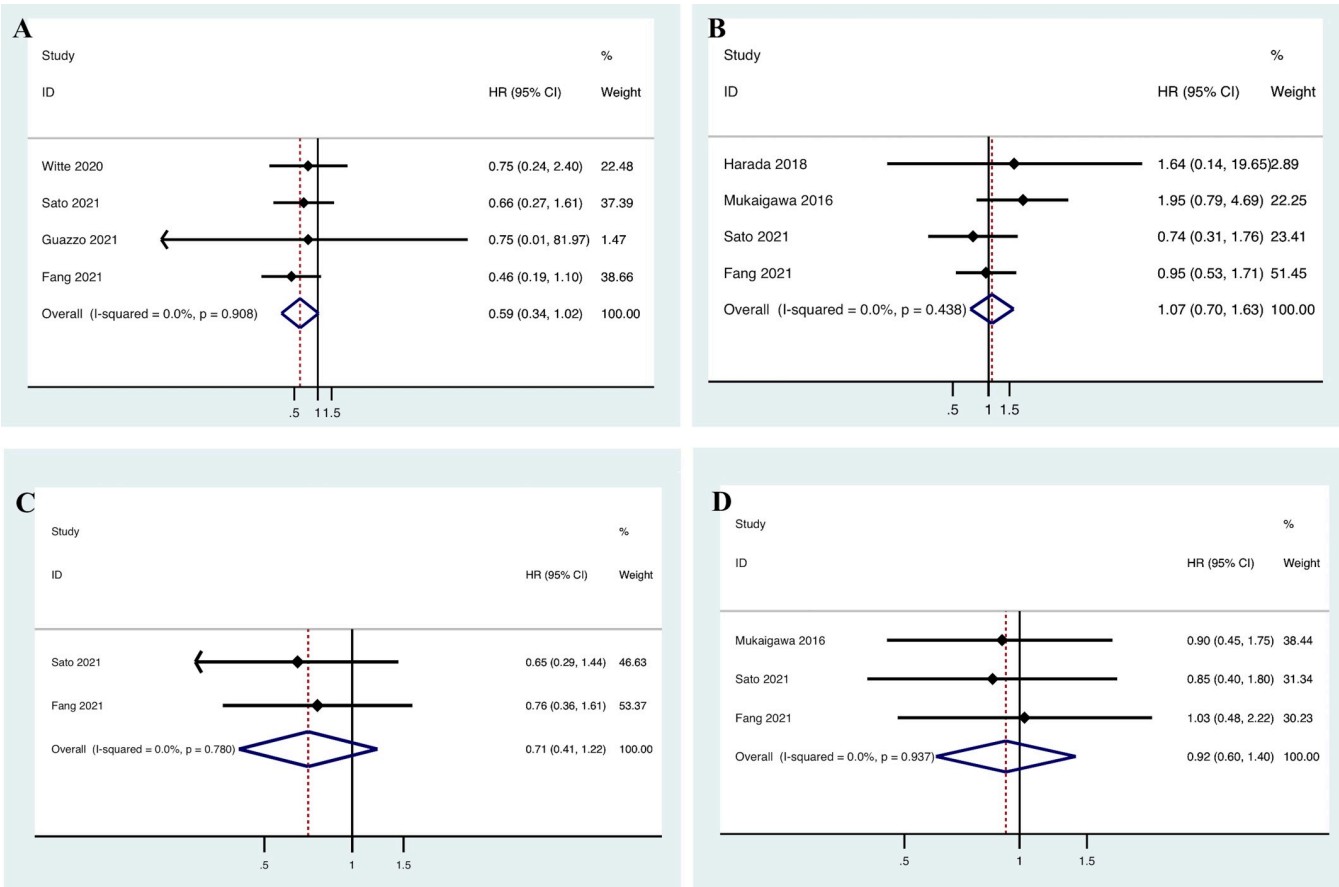

**Fig 4.** Meta-analysis of the prognostic role of PD-L1 expression for the overall survival (OS) and the disease-free survival (DFS) in salivary gland carcinoma (SGC) based on (A, C) combined positivity score (CPS); and (B, D) immune cell (IC). HR, hazard ratio; CI: confidence interval.

presented data with PD-L1 expression based on IC score and DFS [16, 44, 46]. The heterogeneity was not significant ($I^2$ = 0.0%, $p$ = 0.937); therefore, a fixed-effects model was adopted. As shown in Fig 4D, the pooled data demonstrated that there was no association between IC score and DFS of SGC patients (n = 3, HR = 0.921, 95% CI = 0.604–1.403, $P$ = 0.701).

**Association between PD-L1 expression and PFS of SGC patients.** Data on PD-L1 expression and PFS were retrieved from two included studies involving 178 patients [42, 47]. Because of significant heterogeneity ($I^2$ = 66.8%, $p$ = 0.083), a random-effects model was applied. The pooled HR indicated that the correlation between PD-L1 expression and PFS was not significant (n = 2, HR = 1.317, 95% CI = 0.507–3.420, $P$ = 0.572; Fig 5A).

**Association between PD-L1 expression and DSS of SGC patients.** The DSS HRs were derived from two investigations including 157 patients. The heterogeneity was not significant ($I^2$ = 0.0%, $P$ = 0.475); therefore, a fixed-effects model was applied. The pooled HR indicated that the correlation between PD-L1 expression and DSS was not significant (n = 2, HR = 3.567, 95% CI = 0.682–18.664, $P$ = 0.132; Fig 5B).

## Sensitivity analysis

The sensitivity analyses were conducted to evaluate the reliability and stability of the results by removing each study one at a time. As shown in S1 Fig, the pooled ORs for gender, age, T

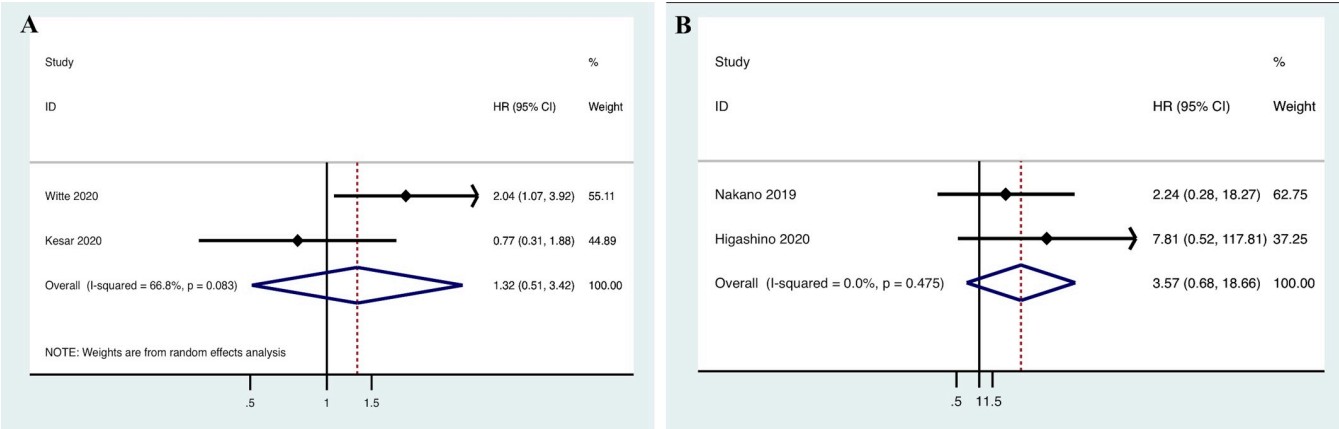

**Fig 5.** Forest plot for the association between programmed cell death ligand 1 (PD-L1) expression and (A) the progression-free survival (PFS); and (B) the disease-specific survival (DSS) of salivary gland carcinoma (SGC) patients. HR, hazard ratio; CI: confidence interval.

stage, N stage, pathological grade, and subtype (non-ACC&ACC) were highly credible. And the sensitivity analyses of the association of PD-L1 and OS and DFS were also performed, and the results of the sensitivity analysis demonstrated high credibility of the pooled HRs as well (S1 Fig). Because of the limited number of studies, we failed to conduct the sensitivity analyses in PFS and DSS. For the same reason, we failed to conduct the publication bias of survival outcomes.

## Discussion

Because of the highly heterogeneous nature and variable clinical course of SGC, patients' survival outcomes are hard to predict. The PD-L1/PD-1 signaling pathway is an important component of tumor immunosuppression, which can inhibit the activation of T lymphocytes and enhance the immune tolerance of tumor cells, thereby achieving tumor immune escape [48, 49]. Recent studies showed that pembrolizumab demonstrated promising antitumor activity and a manageable safety profile in patients with advanced, PD-L1positive SGC [50, 51]. Other recent studies showed the effectiveness and safety of nivolumab for the treatment of SGC [26, 52]. All of these findings suggest that immune checkpoint inhibitors have clinical efficacy in patients with SGC. However, there is yet insufficient evidence to demonstrate a clear predictive role for PD-1/PD-L1 blockade therapy in SGC. This meta-analysis comprehensively and systematically assessed the relationship of PD-L1 expression with clinicopathological features and prognosis in patients with SGC.

In our analysis of clinicopathological features, it was found that men or elder patients with SGC having late tumor stage, lymph node metastasis, high pathological differentiation, or non-ACC subtype were more likely to exhibit high PD-L1 expression. There were no apparent relationships between positive PD-L1 expression and clinical stage, PNI, VI, surgical margin, or treatment failure. A recent meta-analysis performed by Tang et al. [53], included 886 participants and among which 6 studies, demonstrated that PD-L1 was highly expressed in female patients with oral squamous cell carcinoma (OSCC). In the same study, the author found that PD-L1 overexpression was associated with younger age in patients with oropharyngeal squamous cell carcinoma (OPSCC) according to 2 studies that included 212 patients. These findings have contradicted the results of our study. This study proposed that maybe the variant microenvironment of different cancer types and different cutoff values of age among studies

contribute to the differences. The molecular mechanism that accounts for this difference remains to be uncovered. Further studies are needed for investigating these.

He et al. [54] reviewed the PD-L1 expression of 2291 OSCC patients from 17 studies, and the results revealed that positive PD-L1 expression was significantly related to N stage. In the same meta-analysis, which included 18 researches with 2396 OSCC and OPSCC patients, the result showed that the positive rate in patients with late T stage was higher than that of early T stage based on 18 studies including 2396 OSCC and OPSCC patients; however, the difference was not statistically significant. Another meta-analysis that included 1501 OSCC and OPSCC patients from 12 studies examined the correlations between lymph node metastases and PD-L1 expression. The results indicated that there is a tendency between lymph node metastasis and increased PD-L1 expression [53]. The above meta-analysis both demonstrated that positive PD-L1 expression was associated with higher pathological grades. These findings were consistent with the results of our present study of the association between PD-L1 expression and T stage, N stage, and pathological grade in SGC patients.

Immune cell infiltration differs among SGC subtypes, Linxweiler et al. [55] demonstrate that levels of immune infiltration of SGCs were associated with mutation- and fusion-derived neoantigens. They discovered that salivary duct carcinomas (SDC) had significant levels of immune infiltrate, higher levels of T-cell dysfunction and mutational load. ACCs, on the other hand, have an immune-excluded microenvironment and low mutational load. Mucoepidermoid carcinomas (MEC) exhibited both immune-low and immune-high characteristics. Our result of positive PD-L1 expression in the non-ACC subtype is in line with data from Dou et al. [39] and Mosconi et al. [33] who focused on the ACC subtype and demonstrated nearly no PD-L1 expression in ACC. Mosconi et al. observed that immune tolerance phenotype tumors are accompanied by low or no PD-L1 expression in ACC [39]. Maybe the impaired IFN-g response signal transduction in tumor cells could explain these observations to some extent [56]. A recent randomized phase 2 study [57] confirmed low levels of PD-L1 expression and CD8 infiltrating T-cells in patients with recurrent or metastatic ACC. A total of 20 patients in this study underwent pembrolizumab with or without radiation. The results showed that stable disease was more common in patients with demonstrable PD-L1 expression.

Pathological subtypes may influence the prognosis of SGC. For MEC, the OS rate of high-grade and low/intermediate-grade MEC at 5 years was 52% and 83% according to Chen et al. [58]. A recent study conducted by Chan et al. [59] indicated that treatment for MEC with adequate surgical resection and adjuvant radiation therapy yields excellent survival (5-year OS rate of 93.6%), irrespective of pathologic grade. The prognosis of ACC and acinic cell carcinoma (AcCC) is considered favorable as the OS rate at 5 years approaches 90% [60] and 97.2% [61], whereas 5-year OS was poor for carcinoma ex pleomorphic adenoma (Ca-ex-PA) and SDC at 68.5% [62] and 47.9% [63]. In our meta-analysis, three studies presented the data on the influence of pathological subtypes on survival. In the study by Fang 2021, high-grade MEC has been reported as not being an independent prognostic factor for OS or DFS [46]. In the study by Mukaigawa 2016, the histological type was not a significant predictive factor for poor OS, but Ca-ex-PA, SDC, and adenocarcinoma-not otherwise specified (AcNOS) presented worse 5-year OS compared to ACC according to univariate analysis [16]. According to their pathological subtype, Kesar et al. classified the samples as aggressive (ACC, AcNOS, SDC, and high-grade MEC) or non-aggressive (AcCC and low-and intermediate-grade MEC). In univariate analysis, the non-aggressive subtype characteristic was associated with a significantly improved OS and PFS. However, in multivariate analyses, aggressive histopathological subtype characteristic was not an independent prognostic variable [42].

The site of the primary tumor was also an important predictor of outcome in SGC patients. Park et al. investigated the survival outcome of 108 intermediate-grade SGC patients in their

study, from which the most common histologic type was MEC and ACC; the result suggested that a non-parotid primary site is an independent prognostic factor for poor RFS [64]. For ACC and SDC, patients with parotid gland tumors demonstrated more favorable survival outcomes [65, 66], And a primary site in the submandibular gland was an independent prognostic factor for worse survival outcomes [63, 67]. The tumor site (major and minor salivary gland) did not affect survival in Ca-ex-PA patients [68]. In our meta-analysis, two studies presented the data on the influence of primary tumor sites on survival outcomes. In the study by Mukaigawa 2016, the primary site of the major or minor salivary gland was not a predicted factor for DFS or OS [16]. According to the study by Fang 2021, the main histological types were high-grade MEC and SDC, a primary site in the parotid gland that was an independent prognostic factor for worse DFS and OS in patients with high-grade SGC [46]. The above conflicting results should be attributed to the heterogeneity of pathological subtypes and the grade of included cases. When investigating the prognosis of SGC, various risk factors, such as pathological subtypes, grades, and primary tumor location, should be considered simultaneously.

PD-L1 expression has been studied for its prognostic value in head and neck cancer (HNC) in various meta-analysis studies. Li et al. [69] analyzed the survival data of 2869 HNC patients, this meta-analysis showed that the positive expression of PD-L1 was associated with shorter OS and shorter DFS in patients from Asian countries/regions. They also discovered that positive PD-L1 expression was associated with a poor OS in patients with OSCC. This finding is consistent with a recent meta-analysis, which based on data from 26 studies involving 2532 cases, this study showed that PD-L1 overexpression could predict worse DSS and DFS in patients with OSCC [70]. Nevertheless, Yang et al. [71] analyzed survival data of 285 patients with advanced HNSCC patients in their meta-analysis; they found that patients with positive PD-L1 expression showed an improved PFS when compared to those with negative PD-L1 expression. Patel et al. [72] investigated survival outcome and tumor response with 1007 HNSCC patients from twelve studies, they concluded that patients expressing PD-L1 may have a better tumor response and OS. Jia et al. [73] reviewed 52 studies with 7127 HNC patients, they demonstrated that higher expression of PD-L1 correlated with better PFS. These investigations discovered that patients with PD-L1 expression may respond better to PD-L1 inhibitor therapy. Our study examined the survival data of 952 SGC patients from 10 cohort studies, and the findings revealed that PD-L1 expression was significantly associated with SGC patients' OS and DFS. However, there was no significant correlation between PD-L1 expression and PFS or DSS of SGC patients.

According to the findings, positive PD-L1 expression may predict a poor prognosis in SGC patients. These findings would help to establish the rationale for the therapeutic potential of targeting the PD-1/PD-L1 pathway in SGC patients. Moreover, our Meta-analysis indicated that SGC with a high-pathological grade presented with high PD-L1 expression, and high PD-L1 expression indicate a poor prognosis. These results support the hypothesis that SGC with positive PD-L1 expression is more aggressive and has a poor prognosis, which is consistent with previous research. As a result, our findings may have clinical implications for guiding the optimal clinical use of PD-1/PD-L1 inhibitors in these patients. In this meta-analysis, we did not detect the relationship between PFS and positive PD-L1 expression in SGC patients. PFS is mainly used in patients with advanced staged or recurrent/metastatic cancer. However, the majority of our included studies were based on primary tumors, and only two studies included both primary and recurrent tumors [42, 47]. Two studies investigated DSS and PD-L1 expression in SGC patients [41, 45], and the pooled results presented no association between PD-L1 expression and DSS. These results may be attributed to the limited number of included studies, in which change in any included study should influence the pooled results.

Because there were fewer studies included, the link with PFS/DSS was not dominant. More research is required to clarify the results.

Different scores have been introduced for the assessment of PD-L1 expression. TPS/PD-L1 positivity has been widely used to assess the prognostic value for HNC patients in meta-analysis. He et al. [54] found that high PD-L1 expression in OSCC was not related to OS, Tang et al. [53] demonstrated that the higher PD-L1 expression in tumor cells indicated a better DFS of OPSCC, and Yang et al. [71] claimed that the positive PD-L1 expression might predict better PFS in patients with advanced HNSCC. Furthermore, in a randomized phase 3 clinical study for recurrent/metastatic HNSCC, TPS-and CPS-PD-L1 positive was employed as a predictor of responses to PD-1/PD-L1 inhibitors [74, 75]. However, the significance of PD-L1 expression according to these scores in SGC remains unclear. One published clinical trial data for SGC set PD-L1 expression on ≥1% of tumor or stroma cells as positive expression. This study included 26 patients treated with pembrolizumab for unresectable/metastatic PD-L1 positive SGC. The objective response rate was 12%, and 12 patients (46%) had stable disease [50]. The other phase 2 study used ≥1% of tumor cells as positive PD-L1 expression in their study. This study included recurrent or metastatic ACC patients, and 60% patients underwent pembrolizumab treatment presented with stable disease [57]. In patients with advanced PD-L1 positive SGC, a PD-1 inhibitor with a manageable safety profile demonstrated potential antitumor efficacy. Our meta-analysis demonstrated that positive PD-L1 expression based on TPS, rather than CPS or IC scores indicate an important prognostic role and should be a better promising target for immunooncologic treatment.

There were no subgroup parameters that contributed to the study's heterogeneity. Our subgroup analysis indicated that >1% may be a better cut-off value for PD-L1 positivity in SGC patients. Tissue microarray can obtain high throughput information of histomorphology and protein expression on a single section with a small tissue core. Whereas, for the reason of inter-tumoral heterogeneity, a small tissue sample does not represent the overall expression of PD-L1 [76]. Our study suggests that the whole tissue section should be the optimal tissue for the investigation of PD-L1 expression in SGC patients.

Some limitations of this meta-analysis must be considered when interpreting the results. First, studies on heterogeneity limit the validity of outcomes, which might be due to the diversity of pathological subtypes, intratumoral heterogeneity of SGC, different primary cancer sites and stages, and different treatment regimens. A more comprehensive analysis, such as individual participant data meta-analysis with large sample size studies, should be performed to improve the quality of data and confirm the findings. Second, although the prevalence of SGC is relatively low, the sample size with 10 studies that included 952 patients were not favorable for effective pooled analyses, high-quality multicenter clinical trials should be conducted to verify our conclusions. Third, all included studies were retrospective, which presented a relatively inferior level of evidence; data from prospective studies will be needed to gain certainty in the evidence.

## Conclusions

In summary, our meta-analysis indicated that positive PD-L1 expression was significantly associated with certain clinicopathological features, including gender, age, T stage, lymph node metastasis, and pathological grade and type in patients with SGC. Higher expression of PD-L1 was associated with poor OS and DFS of SGC patients. Thus, our findings imply that elevated PD-L1 expression may be a significant predictor of poor prognosis in SGC patients. Large-scale, prospective studies are still needed to confirm the findings. The assessment of PD-L1 expression may aid in the personalized management of SGC.

## Supporting information

**S1 Table. Search strategy of database.**
(DOCX)

**S2 Table. PRISMA checklist.**
(DOCX)

**S3 Table. Clinicopathological features of included studies.**
(XLSX)

**S4 Table. Survival parameters of included studies.**
(XLSX)

**S1 Fig. Sensitivity analysis.**
(DOCX)

## Acknowledgments

We would like to thank the researchers and study participants for their contributions.

## Author Contributions

**Conceptualization:** Limeng Wu, Canhua Jiang.

**Data curation:** Limeng Wu, Zhihui Zhu, Yao Sun, Tao Zhang.

**Formal analysis:** Limeng Wu, Canhua Jiang.

**Methodology:** Limeng Wu, Canhua Jiang, Zhihui Zhu.

**Software:** Limeng Wu.

**Supervision:** Canhua Jiang, Tao Zhang.

**Writing – original draft:** Limeng Wu, Tao Zhang.

**Writing – review & editing:** Limeng Wu, Canhua Jiang, Zhihui Zhu, Tao Zhang.

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
