## [Decision Letter · Decision Letter 0]

15 May 2022

PONE-D-22-01399Prognostic role of PD-L1 expression in patients with salivary gland carcinoma: a systematic review and meta-analysisPLOS ONE

Dear Dr. Zhang,

Thank you for submitting your manuscript to PLOS ONE. After careful consideration, we feel that it has merit but does not fully meet PLOS ONE’s publication criteria as it currently stands. Therefore, we invite you to submit a revised version of the manuscript that addresses the points raised during the review process.

ACADEMIC EDITOR:

Please review and address the comments from the reviewers.

We look forward to receiving your revised manuscript.

Kind regards,

Andrew Birkeland, M.D.

Academic Editor

PLOS ONE

Journal Requirements:

Additional Editor Comments:

The authors perform a meta-analysis of analyzing the association of PD-L1 expression with specific clinicopathological features and prognosis outcomes in salivary gland carcinoma. Please review and address the thoughtful comments from both reviewers.

Reviewers' comments:

Reviewer's Responses to Questions

**Comments to the Author**

1. Is the manuscript technically sound, and do the data support the conclusions?

Reviewer #1: Yes

Reviewer #2: Yes

2. Has the statistical analysis been performed appropriately and rigorously? 

Reviewer #1: Yes

Reviewer #2: Yes

3. Have the authors made all data underlying the findings in their manuscript fully available?

Reviewer #1: Yes

Reviewer #2: Yes

4. Is the manuscript presented in an intelligible fashion and written in standard English?

Reviewer #1: Yes

Reviewer #2: Yes

5. Review Comments to the Author

Reviewer #1: The authors have contributed a unique manuscript investigating the prognostic role of PD-L1 expression on patients with salivary gland carcinoma. This manuscript has been registered a priori to PROSPERO, and firmly adheres to PRISMA guidelines. Although there are certain limitations, this paper provides valuable insights into the future management of PD-L1 positive salivary gland carcinomas.

Key limitations include that this review is based on

• Lines 114-117 (Data extraction and quality assessment)

Please clarify if the third reviewer resolved conflicts during the full-text examination ONLY or also for the title/abstract screen. How many authors participated in the full-text assessment? (I assume the prior two, but this should be clearly expressed.)

• Lines 402-406 (Discussion)

The authors repeat this finding from the results in the discussion: “significantly associated with SGC patients' OS and DFS… no significant correlation between PD-L1 expression and DFS or DSS of SGC patients”

First, should the second DFS be PFS? Please ensure the validity/integrity of the data/findings reported prior to resubmission.

What rationale do the authors suggest for these findings? How do the statistically differences between OS/DFS and the nonsignificant differences between PFS/DSS matter clinically?

• Lines 438-446 (Limitations)

This systematic review is based on studies conducted with retrospective data (inferior level of evidence). Please elaborate on this in the limitations section.

• Discussion

Although the authors mention that the heterogeneity between subtypes/treatment limit outcomes in the limitations section, this topic should be further elaborated on.

For instance, significant differences in prognosis exist between subtype (mucoepidermoid carcinoma vs. acinic cell carcinoma), or primary sites (submandibular vs parotid). Further discussion on this is suggested.

Reviewer #2: The authors present a meta-analysis of analyzing the association of PD-L1 expression with specific clinicopathological features and prognosis outcomes in salivary gland carcinoma. They identified 10 independent cohort studies that met their inclusion criteria. The evidence shows a significant association between PD-L1 expression and male gender, older age, T stage, positive N, higher pathological grade, and non- adenoid cystic carcinoma (ACC) subtype. Additionally, they showed a significance between PD-L1 expression and overall survival in the <100 patient subgroup (5 studies) and disease-free survival with pooled hazard ratio data (5 studies).

The study is well written, and the authors used appropriate statistical modeling to account for between study variation.

The study was a literature review, thus no IRB or consent issues

I have two questions/comments for the authors:

1. For the clinicopathological features, how did you define Old or Young. The cutoff values for age listed under table 2 are not clear for what you used to define this criteria.

2. Could you expand further on the meta-regression used for the survival outcomes in your discussion.

Minor points:

1. The 4th and 5th sentences under “Risk of bias in studies” are unclear. 4th sentence makes no sense, and in the 5th sentence do you mean “in-complete follow-up, or evidence of participants missing”?

2. In the first sentence under “Association between PD-L1 expression and OS of SGC patients” do you mean “extracted from seven studies” versus “included research”

3. I am also confused by your use of “with 2 types of research” (line 350) and “which included 18 researches” (Line 360)

4. Space needed between PD-L1 and positive (line 424)

6. PLOS authors have the option to publish the peer review history of their article (what does this mean?). If published, this will include your full peer review and any attached files.

Reviewer #1: No

Reviewer #2: No

---

## [Author Response · Author response to Decision Letter 0]

12 Jun 2022

Response to Reviewers

Dear Dr. Birkeland,

Thank you for your letter and for the reviewers’ comments concerning our manuscript entitled “Prognostic role of PD-L1 expression in patients with salivary gland carcinoma: a systematic review and meta-analysis” (ID: PONE-D-22-01399). Those comments are all valuable and very helpful for revising and improving our paper, as well as the important guiding significance to our researches. We have studied comments and proofread the manuscript carefully and have made correction which we hope meet with approval. Revised portion are highlighted in different colors in the paper. Here below is our point by point description on revision according to the reviewers’ comments:

Reviewer #1: 

The authors have contributed a unique manuscript investigating the prognostic role of PD-L1 expression on patients with salivary gland carcinoma. This manuscript has been registered a priori to PROSPERO, and firmly adheres to PRISMA guidelines. Although there are certain limitations, this paper provides valuable insights into the future management of PD-L1 positive salivary gland carcinomas.

Key limitations include that this review is based on

1) Comment: • Lines 114-117 (Data extraction and quality assessment)

Please clarify if the third reviewer resolved conflicts during the full-text examination ONLY or also for the title/abstract screen. How many authors participated in the full-text assessment? (I assume the prior two, but this should be clearly expressed.)

Response: We have clarified the number of reviewers for the title/abstract screen and full-text examination in the Data extraction and quality assessment part in Materials and Methods section. (lines 115-116, highlighted in yellow)

2) Comment: • Lines 402-406 (Discussion)

The authors repeat this finding from the results in the discussion: “significantly associated with SGC patients' OS and DFS… no significant correlation between PD-L1 expression and DFS or DSS of SGC patients”

First, should the second DFS be PFS? Please ensure the validity/integrity of the data/findings reported prior to resubmission.

What rationale do the authors suggest for these findings? How do the statistically differences between OS/DFS and the nonsignificant differences between PFS/DSS matter clinically?

Response: The second DFS should be PFS. The “DFS” was changed to “PFS” (line 466, highlighted in yellow). We have added the rationale for the findings of our study in Discussion section. The result of statistically differences between OS/DFS and the nonsignificant differences between PFS/DSS were also discussed in the same paragraph. (lines 470-487, highlighted in yellow)

3) Comment:• Lines 438-446 (Limitations)

This systematic review is based on studies conducted with retrospective data (inferior level of evidence). Please elaborate on this in the limitations section.

Response: We have discussed the influence of retrospective data for the outcome for our study in the last paragraph (limitations section) of discussion. (lines 527-529, highlighted in yellow)

4) Comment:• Discussion

Although the authors mention that the heterogeneity between subtypes/treatment limit outcomes in the limitations section, this topic should be further elaborated on.

For instance, significant differences in prognosis exist between subtype (mucoepidermoid carcinoma vs. acinic cell carcinoma), or primary sites (submandibular vs parotid). Further discussion on this is suggested.

Response: The influence of pathological subtype and/or primary site on SGC patient prognosis was discussed in Discussion section. (lines 409-446, highlighted in yellow)

REVIEWER #2

The authors present a meta-analysis of analyzing the association of PD-L1 expression with specific clinicopathological features and prognosis outcomes in salivary gland carcinoma. They identified 10 independent cohort studies that met their inclusion criteria. The evidence shows a significant association between PD-L1 expression and male gender, older age, T stage, positive N, higher pathological grade, and non- adenoid cystic carcinoma (ACC) subtype. Additionally, they showed a significance between PD-L1 expression and overall survival in the <100 patient subgroup (5 studies) and disease-free survival with pooled hazard ratio data (5 studies).

The study is well written, and the authors used appropriate statistical modeling to account for between study variation.

The study was a literature review, thus no IRB or consent issues

I have two questions/comments for the authors:

1) Comment: 1. For the clinicopathological features, how did you define Old or Young. The cutoff values for age listed under table 2 are not clear for what you used to define this criteria.

Response: The cutoﬀ value of age was based on the median age of the included studies. We have added the explanation of the cutoff value in Table 2 footnotes. (line 230, highlighted in green)

2) Comment: 2. Could you expand further on the meta-regression used for the survival outcomes in your discussion. 

Response: We found the heterogeneity among studies was not significant when we investigated the association between PD-L1 expression and OS of SGC patients. The meta-regression analysis also indicated that no subgroup significantly contributes to heterogeneity of OS, so we did not further expand the meta-regression used for the survival outcomes in discussion.

Minor points:

3) Comment: 1. The 4th and 5th sentences under “Risk of bias in studies” are unclear. 4th sentence makes no sense, and in the 5th sentence do you mean “in-complete follow-up, or evidence of participants missing”? 

Response: We have re-written and re-organized the 4th and 5th sentences under “Risk of bias in studies” according to the reviewer’s comments. (lines 236-241, highlighted in green)

4) Comment: 2. In the first sentence under “Association between PD-L1 expression and OS of SGC patients” do you mean “extracted from seven studies” versus “included research” 

Response: We have changed “included research” to “studies” in the first sentence under “Association between PD-L1 expression and OS of SGC patients”. (line 266, highlighted in green)

5) Comment: 3. I am also confused by your use of “with 2 types of research” (line 350) and “which included 18 researches” (Line 360)

 Response: We have re-written these sentences according to the reviewer’s comments. (lines 366-369; lines 377-379, highlighted in green)

6) Comment: 4. Space needed between PD-L1 and positive (line 424)

Response: We have added space between “PD-L1” and “positive”. (line 504, highlighted in green)

Other changes (Highlighted in blue in the revised manuscript):

1) “Materials and methods-Data extraction and quality assessment”: 2nd paragraph, 1st sentence, changed to: “Patient survival outcomes included…”.

2) “Results- Sensitivity analysis”: 2nd sentence, changed to: “As shown in S1 Fig, the pooled… were highly credible”.

3) Reformatted Table1-3.

4) Added “Acknowledgments” section. 

5) File “S1 Fig. Sensitivity analysis”- figure legend for S1 Fig.: changed to “…(F) Subtype (non-ACC&ACC).”

We tried our best to improve the manuscript. All the changes indicated above are in the revised manuscript.

We appreciate for Editors/Reviewers’ warm work earnestly, and hope that the correction will meet with approval.

Once again, thank you and all the reviewers for the comments and suggestions.

Sincerely yours,

Tao Zhang

---

## [Decision Letter · Decision Letter 1]

13 Jul 2022

Prognostic role of PD-L1 expression in patients with salivary gland carcinoma: a systematic review and meta-analysis

PONE-D-22-01399R1

Dear Dr. Zhang,

We’re pleased to inform you that your manuscript has been judged scientifically suitable for publication and will be formally accepted for publication once it meets all outstanding technical requirements.

Kind regards,

Andrew Birkeland, M.D.

Academic Editor

PLOS ONE

Additional Editor Comments (optional):

The authors have revised their manuscript and responded to the reviewers' comments appropriately.

Reviewers' comments:

Reviewer's Responses to Questions

**Comments to the Author**

1. If the authors have adequately addressed your comments raised in a previous round of review and you feel that this manuscript is now acceptable for publication, you may indicate that here to bypass the “Comments to the Author” section, enter your conflict of interest statement in the “Confidential to Editor” section, and submit your "Accept" recommendation.

Reviewer #1: All comments have been addressed

Reviewer #2: All comments have been addressed

2. Is the manuscript technically sound, and do the data support the conclusions?

Reviewer #1: Yes

Reviewer #2: Yes

3. Has the statistical analysis been performed appropriately and rigorously? 

Reviewer #1: Yes

Reviewer #2: Yes

4. Have the authors made all data underlying the findings in their manuscript fully available?

Reviewer #1: Yes

Reviewer #2: Yes

5. Is the manuscript presented in an intelligible fashion and written in standard English?

Reviewer #1: Yes

Reviewer #2: Yes

6. Review Comments to the Author

Reviewer #1: The authors have addressed all concerns appropriately. The reviewer would like to thank the authors for their submission!

Reviewer #2: (No Response)

7. PLOS authors have the option to publish the peer review history of their article (what does this mean?). If published, this will include your full peer review and any attached files.

Reviewer #1: No

Reviewer #2: No

---

## [Editor Report · Acceptance letter]

15 Jul 2022

PONE-D-22-01399R1 

Prognostic role of PD-L1 expression in patients with salivary gland carcinoma: a systematic review and meta-analysis 

Dear Dr. Zhang:

I'm pleased to inform you that your manuscript has been deemed suitable for publication in PLOS ONE. Congratulations! Your manuscript is now with our production department. 

Kind regards, 

on behalf of

Dr. Andrew Birkeland 

Academic Editor

PLOS ONE